# Housing Affordability of Private Rental Apartments According to Room Type in Osaka Prefecture

Mikio Yoshida  and Haruka Kato *

Department of Housing and Environmental Design, Graduate School of Human Life Science, Osaka Metropolitan University, Osaka 5588585, Japan; si22929h@st.omu.ac.jp
* Correspondence: haruka-kato@omu.ac.jp; Tel.: +81-6-6605-2823

**Abstract:** Housing poverty was already a social problem in Japan before the COVID-19 pandemic. The research questions of this study were as follows: How many private rental apartments that are affordable for low-income groups exist in the real estate market? Additionally, do these rental apartments have enough rooms? This study aimed to clarify the housing affordability of private rental apartments in Osaka Prefecture according to room type. In this study, we analyzed housing affordability based on room types and housing conditions using a real estate dataset. In conclusion, this study found that housing affordability is problematic in terms of quantity and quality among the private rental apartments for multiple households in Osaka Prefecture. Additionally, it was found that the role of old wooden low-rent housingbuildings has declined as affordable housing. In particular, the total number of two-room and over three-room-type low-rent housing was less than 8000 units, accounting for only 4.2% of all private rental apartments in the real estate market. The distributed supply of low-rent housing has potential risks in maintaining a stable life for low-income groups with multi-person households. Those low-income groups are forced to live in higher-rent housing or one-room-type low-rent housing.

**Keywords:** housing affordability; low-rent housing; private rental apartments; room types; real estate dataset; Osaka Prefecture

## 1. Introduction

### 1.1. Background

Since 2020, the spread of the coronavirus disease 2019 (COVID-19) pandemic has led to a decline in income, especially among low-income groups (LIGs) [1]. For LIGs whose income decreased due to the COVID-19 pandemic, housing costs have put pressure on the cost of living, because housing costs must be paid regularly, and the prices rarely fall.

However, especially among LIGs, housing poverty was already a social problem before the COVID-19 pandemic [2]. In Japan, housing policy is classified as the dualist model [3]. Under this model, the Japanese housing policy attempted to increase home ownership by mortgage system in relation to economic policies as well. However, people's lifestyles have become increasingly more diverse due to the rising divorce rate [4]. Additionally, after the bubble economy collapsed in Japan, the real estate market in some areas stagnated for a long period of time [5]. In addition, the Japanese government has gradually changed to encourage neoliberal policies. These neoliberal policies reduced the number of public rental apartments and increased that of private rental apartments [6]. These factors led to a decrease in the number of people living in owned houses and an increase in the number of people living in rental housing, especially in rental apartments [7]. This suggests that many LIGs have suffered from housing affordability problems.

The research questions of this study are as follows: How many private rental apartments that are affordable for LIGs exist in the real estate market? Additionally, do these rental apartments have enough rooms? Mulliner et al. [8] indicated that the housing affordability issue encompasses more than financial costs of housing and extends to larger

issues of social wellbeing and sustainability. Due to the change in financialization policies in the housing sector, the stock of public rental apartments declined from 2.18 million in 2003 to 1.92 million in 2018 in Japan [9]. Through the revision of the Act on Public Housing in 1996, governments have restricted the income criteria for those who can live in public rental apartments [10]. Since 1999, some public rental apartments have been managed by the Urban Development Corporation, which is a special public organization. The Urban Development Corporation set rental prices similar to those of private rental apartments [11]. Those political changes made the housing sector's largest players change from public to private sectors. One of the leading players in the private sector is the Residential Real Estate Investment Trusts in Japan (residential J-REITs). The residential J-REITs are observed as strongly contributing to the mixed-asset portfolio context in Japan across the portfolio risk spectrum, particularly in a post-GFC context [12]. However, the residential J-REITs might accelerate the excessive real estate investment and increase housing rental prices even for the housing for LIGs.

For the LIGs, the Japanese government attempted to increase the low-rent housing (LRH) available in the real estate market for LIGs through the Housing Safety Net Law, which is policy in the private housing sector. However, there has been little success under this law [6]. Additionally, LIGs have limited access to public rent assistance programs. For example, LIGs cannot obtain long-term support for the Housing Security Benefit [13]. Moreover, LIGs cannot obtain housing-only support under the Public Assistance for Livelihood Protection [14]. These factors have exacerbated housing affordability issues for LIGs in Japan.

### 1.2. Purpose

This study aimed to clarify the housing affordability of private rental apartments in Osaka Prefecture according to room type. Room types are essential indicators that affect not only the gross floor area of apartments, but also the privacy protections for residents and housing prices. In addition, Bangura et al. [15] clarified that housing affordability varies across different sub-markets. For this purpose, we set the rental price criteria that reflect affordability for LIGs. Based on these price criteria, we then analyzed the LRH using a real estate dataset. The results could help to improve housing affordability for the government and real estate companies.

Housing affordability refers to the cost of housing services and shelter for both renters and owner–occupiers relative to individual or household disposable income [16]. Housing affordability refers to the rent-to-income ratio or house-price-to-income ratio [16]. Based on the rent-to-income ratio, we defined LRH as rental apartments with monthly rent under JPY 51,000 (about USD 440). The price of JPY 51,000 was calculated as the reasonable housing cost burden based on the annual income standard of JPY 2,043,999 (about USD 17,600) for tax-exempt households in Osaka City [17]. Households exempt from residential tax include disabled households, elderly people living alone, and single-parent households. We set the appropriate rent-to-income ratio to 30% of monthly income [18]. Although housing affordability can include housing and transportation costs [19], this study only analyzed rental prices. The criterion of rent-to-income ratio being 30% of monthly income is adequate because many previous papers use these criteria for analyzing housing affordability [20,21].

There are no clear criteria for LIG income and rental prices in Japan. Therefore, there are several possible rent standards other than JPY 51,000. For example, the criterion for the Housing Safety Net Law is about JPY 47,000 (USD 405) [22]. The criterion for housing assistance under the Public Assistance Program is between about JPY 40,000 (about USD 345) and about JPY 62,000 (about USD 535), depending on the number of household members [14]. Among these criteria, tax-exempt households are most easily understood. Similar criteria were also used in previous studies [23]. Therefore, JPY 51,000 was determined to be the most appropriate criterion for LRH.

### 1.3. Case Study

This study analyzed Osaka Prefecture to examine housing affordability. This area was selected for three reasons. First, there are many households with LIGs in Osaka Prefecture. In 2019, the ratio of people receiving Public Assistance for Livelihood Protection compared with the total population in Osaka Prefecture was 3.2%, the highest in Japan [24]. Therefore, analyzing Osaka Prefecture is significant for considering housing affordability in Japan.

Second, Osaka Prefecture has reserved LRH in the private sector, which has played a significant role in the housing supply since the postwar period in Japan. Tsuda et al. [25] found that LIGs in the 1980s mostly lived in old wooden LRH in Osaka Prefecture, such as wooden terrace houses (Nagaya) and wooden apartments (Bunka). Tsuda et al. [26] described five characteristics of Nagaya: rental apartments, a gross floor area of about 40–50 m$^2$, a wooden structure, one or two stories, and a terrace-house type. Additionally, Tsuda et al. [26] described five characteristics of Bunka: rental apartments, a gross floor area of about 20 m$^2$, a wooden structure, two stories, and single-corridor type. Tokuono et al. [27] evaluated the Bunka as the origin of private rental apartments in Japan. Koito et al. [28] noted that several Nagaya homes were renovated according to modern lifestyles. Therefore, analyzing Osaka Prefecture is significant for studying the development of this postwar history of LRH in private rental apartments.

Third, Osaka Prefecture lacks LRH, which causes housing affordability issues. This study analyzed households with annual incomes under JPY 2,000,000 (about USD 17,100) who live in private rental apartments in Osaka Prefecture, using the housing and land survey from 2018 [29]. As a result, the number of households that spent less than JPY 50,000 (about USD 425) on rent was found to be 202,600, whereas households that spent more than JPY 50,000 on rent totaled 158,000. This factor suggests that only about 56% of LIGs can affordably live in LRH. Therefore, analyzing Osaka Prefecture is significant for solving the housing affordability issues for LIGs.

### 1.4. Literature Review

The housing affordability problem has been studied from several viewpoints in many countries. In South Africa, Alabi et al. [30] clarified that the housing affordability problem was caused by an increase in the costs of building materials connected to fluctuations in construction costs and the rise in maintenance costs. However, the housing affordability problem was also caused by social problems. In Malaysia, Liu et al. [31] clarified that the housing affordability problem was caused by low household income, high land price, construction costs and compliance costs, mismatches of supply and demand in terms of quantity, the instability of the national economy, low home financing ability, and incomprehensive housing planning. In South Korea, the housing affordability problem has become a political issue. Lee et al. [32] clarified that young people are burdened with housing costs. For young people, housing poverty can lead to health problems such as anxiety, lethargy, and difficulty preventing and treating disease [33]. Due to those problems, many policymakers have worked to improve housing affordability worldwide. The originality of the current paper lies in its analysis of housing affordability in Osaka Prefecture, which has reserved LRH in private rental apartments.

Regarding the housing affordability problem in Japan, Kawata et al. [34] clarified that people who live in private rental apartments tend to bear a higher burden of housing costs in their cost of living. Therefore, Hirayama et al. [35] clarified that LIGs without the support of their parents are burdened with higher housing costs. These studies clarified housing affordability from the demand side, such as homeownership relationships and household types. On the other hand, this study analyzed housing affordability from the supply side using a real estate database. The results provide significant insights into the adequacy of supply to improve housing affordability problems.

On the supply side, Kishioka et al. [23] analyzed the LRH in the Tokyo Metropolitan area. As a result, it was found that LRH in the private sector tends to be low and offers poor housing conditions with wooden structures. Additionally, it was also found that rent

prices tend to be high regardless of low and poor housing conditions. Shiki [36] clarified that LRH in the private sector decreased due to a decline in wooden apartment buildings without bathrooms. Based on previous studies on the supply side, the novelty of this paper is in its analysis of LRH in Osaka Prefecture according to room types, which is one of the sub-markets.

## 2. Materials and Methods

### 2.1. Real Estate Dataset

This study used an at-home dataset registered from 1 January 2018 to 31 December 2018 [37]. This at-home dataset was registered to the At Home Real Estate Information Network operated by At Home Co., Ltd. At Home Co., Ltd. is a major company that provides real estate information services for real estate companies and consumers. More than 50,000 local real estate companies use this dataset service throughout Japan, which is the largest number of users for such a service in the country. This study comprehensively analyzes supply side data which are traded in the private real estate market, although this study could not analyze the demand-side data. There is little public disclosure on actual transaction prices between individuals in Japan. Therefore, the prices in this dataset provide essential indicators for estimating transaction prices. The data are reliable and have been used in previous studies [38].

This study focused on rental apartments to analyze housing affordability. The dataset not only included rental apartments, but also apartments for sale, detached houses for rent, detached houses for sale, vacant land for sale, and rental shops. Among the categories in the dataset, the rental apartments included old wooden LRH such as Nagaya and Bunka.

The dataset contained variables such as the monthly rental price (JPY), gross floor area ($m^2$), floorplan types (nLDK), structure, year of construction, location, latitude/longitude, facilities, building ID, property number, and duplicate processing key. We extracted approximately 790,000 apartments from the dataset. However, the data included duplicates, such as the same properties that were repeatedly listed every month. Ultimately, we extracted approximately 180,000 apartments from about 790,000 duplicates using the variables of building ID, property number, and duplicate processing key. We then analyzed the 180,000 apartments.

### 2.2. Data Analysis

We analyzed housing affordability based on room type and housing conditions. Room type refers to the number of private rooms in the rental apartments. The classifications are one-room types, two-room types, and over-three-room types. Here, the room types are classified using the number "n" in the variants of floorplan types (nLDK). In Japan, "nLDK" indicates a floorplan based on letter combinations of "n", "L", "D", and "K". In nLDK, "L" means living, "D" means dining, "K" means kitchen, and "n" means the number of private rooms. Therefore, floorplan types of 2K, 2DK, and 2LDK are classified as two-room types. Figure 1 shows the sample of architectural plans for one-room types, two-room types, and over-three-room types. Each architectural plan was drawn by M.Y.

Housing condition refers to the home's structure, year of construction, and location. The structure is classified into five types: wooden, steel, reinforced concrete (RC), steel-reinforced concrete (SRC), and other structures. The year of construction is classified into three types: before 1981, 1982–2001, and 2002–2018. This classification means that the criteria were set every 20 years based on 1981. In Japan, where wooden houses are a typical type of housing structure, 20 years is considered the period required for rebuilding or renovating. In 1981, the Building Standard Law in Japan was revised to design buildings to resist strong earthquakes. This policy means that apartments built before 1981 might be destroyed by strong earthquakes. Therefore, apartments built before 1981 need their structures rebuilt to resist strong earthquakes. Additionally, apartments built from 1982 to 2002 need to be renovated.

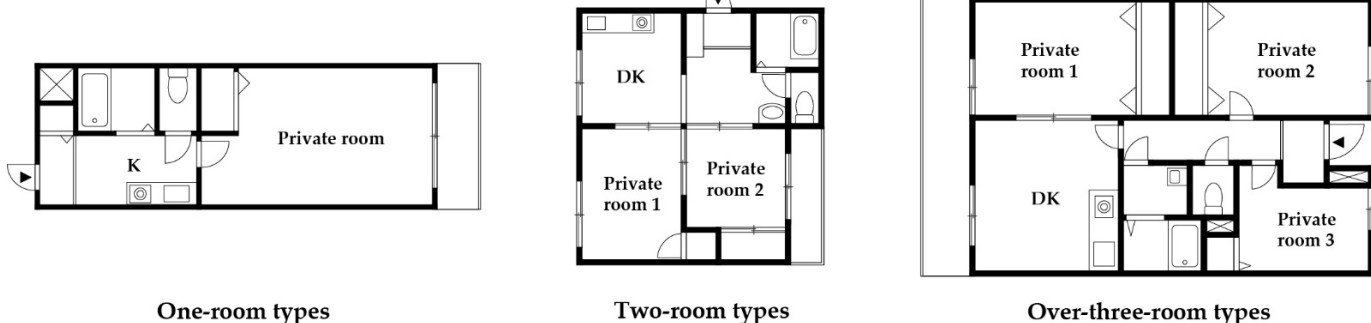

**Figure 1.** Sample of architectural plans of each room type.

From the two aspects of room types and housing conditions, we studied four analyses. First, we present an analysis of the supply of LRH in private rental apartments in Section 3.1. This analysis explains the quantitative sufficiency of LRH.

Second, we present the cross-tabulation of LRH and room types in Section 3.2. Pearson's chi-squared test was used for cross-tabulating. The analysis helps to explain the quantitative sufficiency of LRH from the perspective of room types.

Third, we focus on LRH and present a cross-tabulation of room types and housing conditions in Section 3.3. Pearson's chi-squared test was used for cross-tabulating. In this study, we extracted three quantitative types according to room type based on the analysis. We also extracted the old wooden LRH, such as wooden terrace houses (Nagaya) and wooden apartments (Bunka), which were previously utilized as LRH in Osaka. In the analysis, we defined old wooden LRH, including Nagaya and Bunka, as wooden rental apartments built before 1981 with monthly rent under JPY 51,000. Section 3.3 analyzes the above three housing types and the old wooden LRH in terms of year of construction, building structure, and gross floor areas.

Finally, we analyze the locations of these types based on kernel density estimation, as reported in Section 3.4. In the kernel density estimation analysis, the output cell size was set to 0.0017, and the bandwidth was set to 1000 m. This analysis will help explain the qualitative sufficiency of LRH.

## 3. Results

### 3.1. Supply of Low-Rent Housing

In this section, we analyze the supply of private rental housing using the at-home dataset from 2018 in Osaka Prefecture. Figure 2 shows the supply histogram according to rental price. In Figure 2, the histogram is a normal distribution, with 179,663 rental apartments traded on the real estate market in 2018. Based on the housing and land survey in 2018, there were 439,600 rental apartments. Therefore, 40.9% of rental apartments were traded on the real estate market in 2018. Additionally, Figure 2 shows 64,937 rental apartments with rental prices under JPY 51,000. This result represents 36.1% of total rental apartments in the at-home dataset.

The results were considered based on the housing and land survey in 2018 [9]. The housing and land survey is a comprehensive survey of housing and land in Japan conducted once every five years by the Statistics Bureau in Japan. Therefore, the survey results provide an understanding of rental homes that did not trade in the market. The percentage of households with monthly rent prices under JPY 50,000 and annual incomes of less than JPY 2,000,000 was 29.5% for all populations in Japan. Figure 2 shows that the percentage of private rental apartments with a monthly rent price under JPY 51,000 was 36.1%. Therefore, the balance of LRH supply might not be a problem in the housing market. However, as mentioned in Section 1.3, the number of households that spent more than JPY 50,000 on rent totaled 158,000. Therefore, if these households want to relocate to rental housing

with a monthly rent price under JPY 50,000, the supply of LRH would be insufficient in the market.

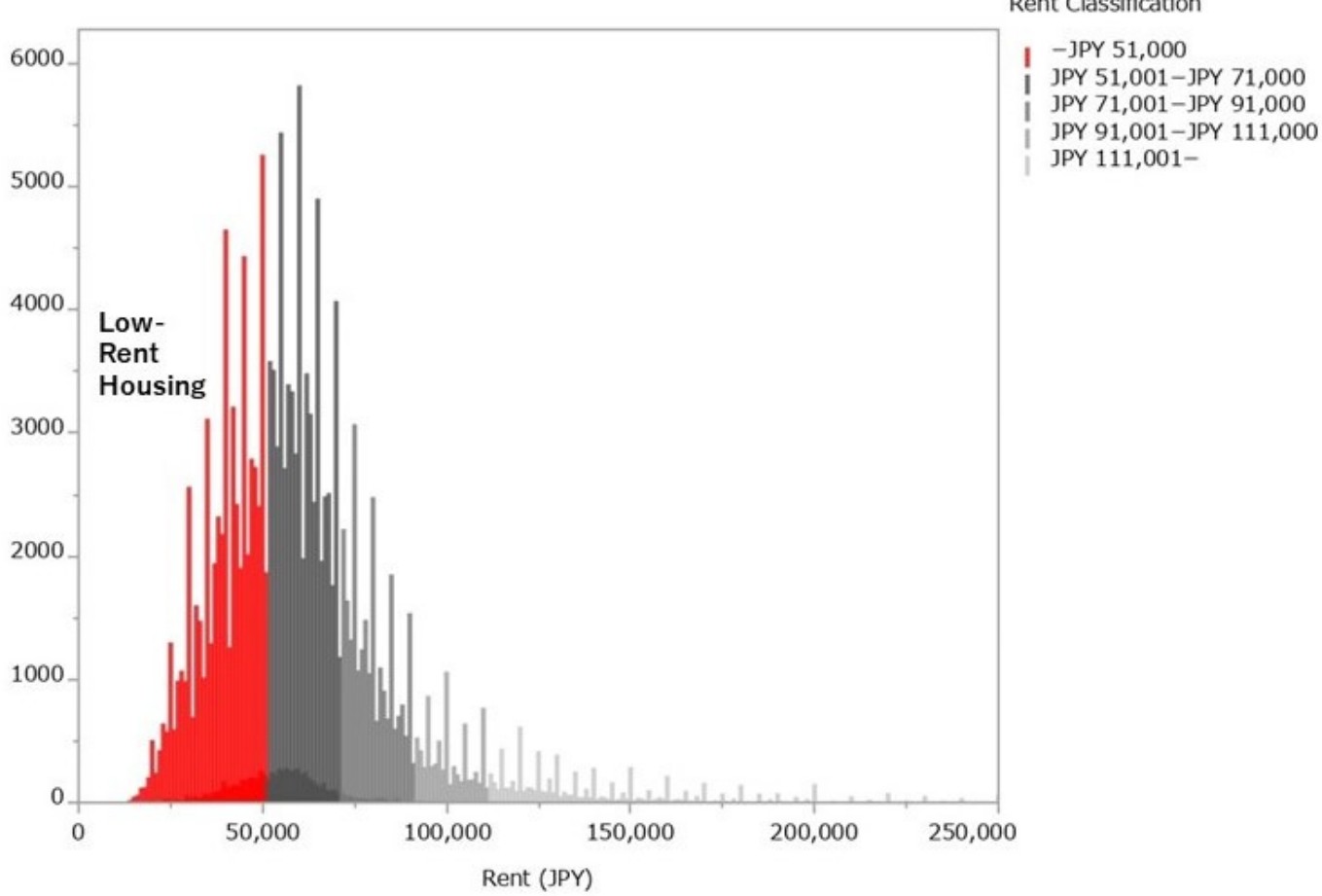

**Figure 2.** Supply of low-rent housing.

There are some important points to note in the above estimations. The first point is that the number of households living in private rental housing not only includes rental apartments, but also detached houses for rent. However, this difference can be ignored, because few detached houses are available for rent. The number of detached houses for rent totals 6262, which is 3.49% of rental apartments using the at-home dataset. The second point is the difference between the standard of JPY 50,000 in the housing and land survey and that of JPY 51,000 in this analysis. However, Figure 2 indicates that this difference can be ignored because the data broadly capture the demand for, and supply of, low-rent housing.

Those results indicate no problem with the supply balance of LRH in the real estate market. However, it was found that the supply volume of LRH is significantly insufficient.

### 3.2. Low-Rent Housing According to Room Type

In this section, we analyze the supply of LRH according to room type. Table 1 presents a cross-tabulation of LRH and room types. In Table 1, Pearson's chi-squared test confirms statistically significant differences at a 0.1% level. Table 1 shows that one-room types are most common, with 120,067 (66.8%) of the total supply. Next, focusing on LRH, most housing is of the one-room type, with 57,350 (88.3%) of the total supply. This result suggests that only about 7587 (11.7%) apartments of LRH have two rooms or over three rooms. Among non-low-rent housing (Non-LRH), 52,009 (45.3%) homes have multiple rooms. This result suggests a lack of LRH with multiple rooms due to the gap in supply for each room type.

**Table 1.** Low-rent housing according to room type.

|  | LRH | Non-LRH | Sum |
|---|---|---|---|
| One-room types | 57,350 (88.3%) | 62,717 (54.7%) | 120,067 (66.8%) |
| Two-room types | 6392 (9.8%) | 30,632 (26.7%) | 37,024 (20.6%) |
| Over-three-room types | 1195 (1.8%) | 21,377 (18.6%) | 22,572 (12.6%) |
| Sum | 64,937 (100%) | 114,726 (100%) | 179,663 (100%) |

$p < 0.001$.

### 3.3. Low-Rent Housing According to Room Type and Housing Condition

In this section, we analyze the LRH according to room type and housing condition. Table 2 shows cross-tabulation using the structure and year of construction according to room type. In Table 2, Pearson's chi-squared test confirms statistically significant differences at a 0.1% level. Based on Table 2, Figure 3 extracts three quantitative types according to room type based on the analysis. Additionally, we extracted one-room and two-room types from among the old wooden LRH. For old wooden LRH, over-three-room types were omitted due to the small amount of data. Figure 3 shows an example picture of a representative exterior. The sources of pictures are the Google Street View. Figure 3 followed the Google Maps/Earth Additional Terms of Service [39]. Additionally, Figure 3 shows the percentage of LRH with certain gross floor area (GFA) values based on the following criteria: 25 m$^2$ for one-room types, 30 and 40 m$^2$ for two-room types, and 40 and 50 m$^2$ for over-three-room types.

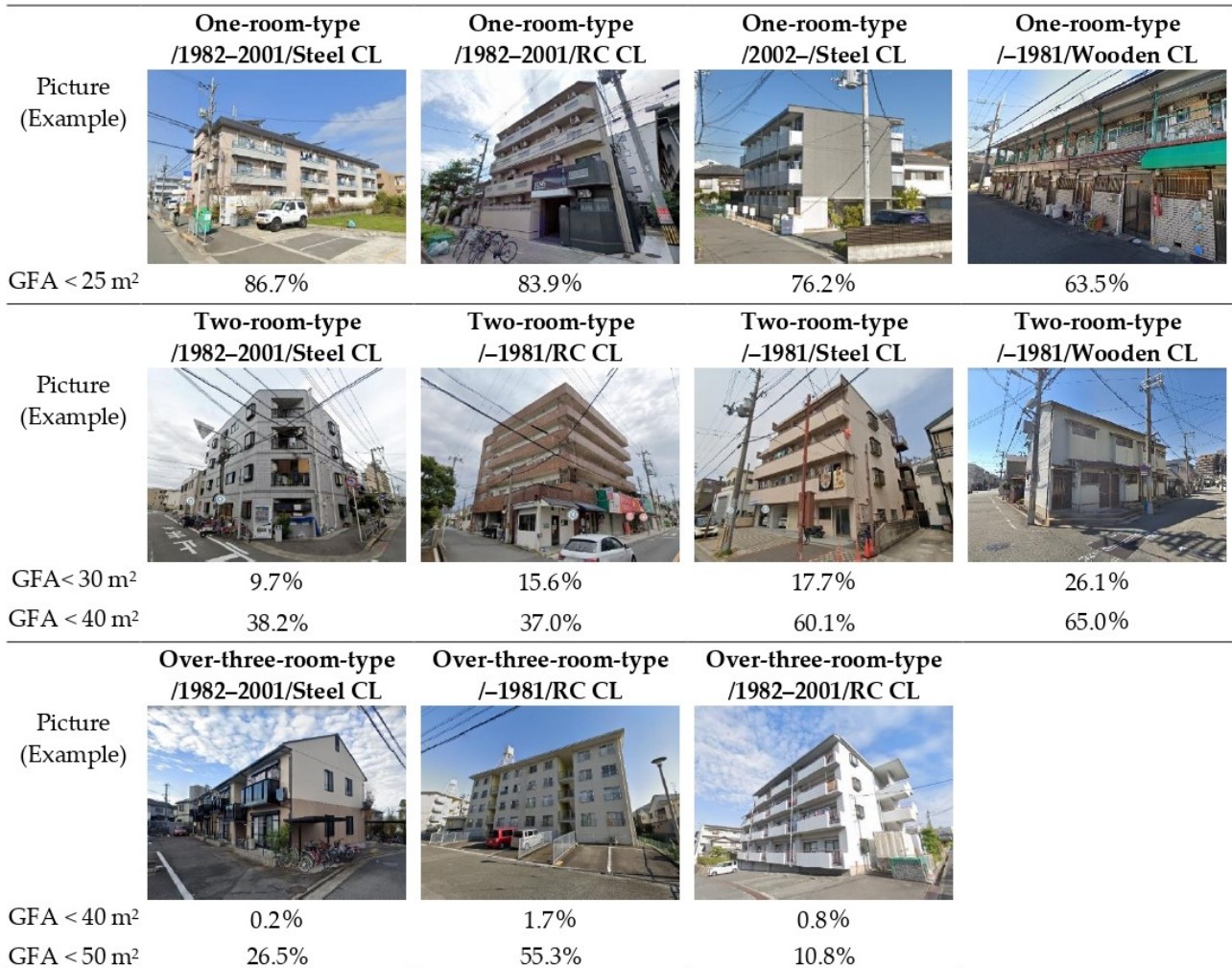

**Figure 3.** Main types of low-rent housing with picture and gross floor area.

**Table 2.** Low-rent housing according to room type and structure.

|  |  | Wooden | Steel | RC | SRC | Others |
|---|---|---|---|---|---|---|
| One-room types | −1981 | 306 | 1357 | 1364 | 483 | 97 |
|  | 1982–2001 | 1964 | 24,196 | 12,248 | 2029 | 1901 |
|  | 2002− | 2150 | 7098 | 1910 | 62 | 185 |
| Two-room types | −1981 | 453 | 951 | 1202 | 99 | 128 |
|  | 1982–2001 | 209 | 2527 | 620 | 18 | 92 |
|  | 2002− | 36 | 52 | 5 | 0 | 0 |
| Over-three-room types | −1981 | 28 | 61 | 360 | 7 | 20 |
|  | 1982–2001 | 33 | 535 | 130 | 12 | 8 |
|  | 2002− | 0 | 1 | 0 | 0 | 0 |

$p < 0.001$.

### 3.3.1. Housing Conditions of One-Room LRH

In this section, we analyze the one-room-type LRH based on Table 2 and Figure 3. We extracted three quantitative types of building structure and year of construction: (1) [one-room-type/1982–2001/Steel Cluster (CL)] (N = 24,196); (2) [one-room-type/1982–2001/RC CL] (N = 12,248); and (3) [one-room-type/2002–/Steel CL] (N = 7098). Additionally, we analyzed [one-room-type/−1981/Wooden CL] (N = 306) as old wooden LRH such as Bunka. After comparing the numbers in the four clusters, we found that the role of the old wooden LRH [one-room-type/−1981/Wooden CL] has declined in the modern era. Additionally, we found that many apartments of one-room-type LRH were not yet severely dilapidated.

For each cluster, we analyzed the ratio of LRH with a GFA under 25 m². The criterion of a GFA under 25 m² is the minimum living area for single households. The percentage of LRH with a GFA under 25 m² was as follows: 86.7% of [one-room-type/1982–2001/Steel CL], 83.9% of [one-room-type/1982–2001/RC CL], 76.2% of [one-room-type/2002–/Steel CL], and 63.5% of [one-room-type/−1981/Wooden CL]. We found that the old wooden LRH had a larger GFA than other LRH. Figure 1 shows that the typical plan of a one-room LRH was a narrow living area for single households.

We found that the physical standards for one-room-type LRH can be evaluated to some extent, such as using large supply and building dilapidation. However, the exceptionally high percentage of narrow LRH needs to be improved.

### 3.3.2. Housing Conditions of Two-Room-Type LRH

In this section, we analyze two-room-type LRH based on Table 2 and Figure 3. For building structure and year of construction, we extracted three quantitative types: (1) [two-room-type/1982–2001/Steel CL] (N = 2527); (2) [two-room-type/−1981/RC CL] (N = 1202); and (3) [two-room-type/−1981/Steel CL] (N = 951). Additionally, we analyzed [two-room-type/−1981/Wooden CL] (N = 453) as old wooden LRH such as Nagaya. After comparing the numbers in the four clusters, we found that the role of the old wooden LRH [two-room-type/−1981/Wooden CL] has declined in the modern era, similar to the results for one-room-type LRH. Additionally, we found that more than 40% of LRH was built under the old Building Standard Law in Japan to resist strong earthquakes, indicating that such houses are seriously deteriorating. In addition, very few LRH units were built after 2001. Subsequently, LRH has decreased rapidly because the supply of LRH has almost stopped.

For each cluster, we analyzed the ratio of LRH with a GFA under 30 and 40 m². A GFA under 30 and 40 m² is the minimum living area criterion for two-person and three-person households, respectively. The ratios of LRH with a GFA under 30 and 40 m² were as follows: 9.7% and 38.2% of [two-room-type/1982–2001/Steel CL], 15.6% and 37.0% of [two-room-type/−1981/RC CL], 17.7% and 60.1% of [two-room-type/−1981/Steel CL], and 26.1% and 65.0% of [two-room-type/−1981/Wooden CL], respectively. It was found that many LRH units have a sufficient GFA for a two-person household. Figure 1 shows that the typical plan of a two-room LRH was a sufficient living area to protect the privacy of two-person and three-person households, including a small child.

We found that two-room-type LRH faces housing affordability problems in physical terms because of the severely deteriorated conditions of such houses and their low supply.

### 3.3.3. Housing Conditions of Over-Three-Room-Type LRH

In this section, we analyze the over-three-room-type LRH based on Table 2 and Figure 3. In terms of building structure and year of construction, we extracted three quantitative types: (1) [over-three-room-type/1982–2001/Steel CL] (N = 535); (2) [over-three-room-type/–1981/RC CL] (N = 360); and (3) [over-three-room-type/1982–2001/RC CL] (N = 130). We found that nearly 40% of LRH units were built under the old Building Standard Law in Japan to resist strong earthquakes, indicating that they are seriously deteriorating. In addition, very few LRH units were built after 2001. Therefore, LRH has decreased rapidly because the supply of LRH has almost stopped, similar to the results for two-room-type LRH.

In each cluster, we analyzed the ratio of LRH with a GFA under 40 and 50 m$^2$. The criteria of a GFA under 40 and 50 m$^2$ is the minimum living area level for four-person and five-person households, respectively. The ratios of LRH with a GFA under 40 and 50 m$^2$ were as follows: 0.2% and 26.5% of [over-three-room-type/1982–2001/Steel CL], 1.7% and 55.3% of [over-three-room-type/–1981/RC CL], and 0.8% and 10.8% of [over-three-room-type/1982–2001/RC CL], respectively. It was found that many LRH units have a sufficient GFA for a four-person household. Figure 1 shows that the typical plan of over-three-room LRH is a sufficient living area to protect the privacy of four-person and five-person households, including children.

We found that three-room-type LRH also faces housing affordability problems in physical terms because of the severely deteriorated condition and low supply of such homes, similar to the results for two-room-type LRH.

### 3.4. Geographic Concentration of LRH

In this section, we analyze the geographic accumulation of the eleven clusters of LRH studied in Section 3.3. Figure 4 shows the clusters based on kernel density estimation. In the kernel density estimation, the output cell size was set to 0.0017, and the bandwidth was set to 1000 m for the analysis. The base map in Figure 4 was open-source Arc GIS PRO, and complies with copyright [40].

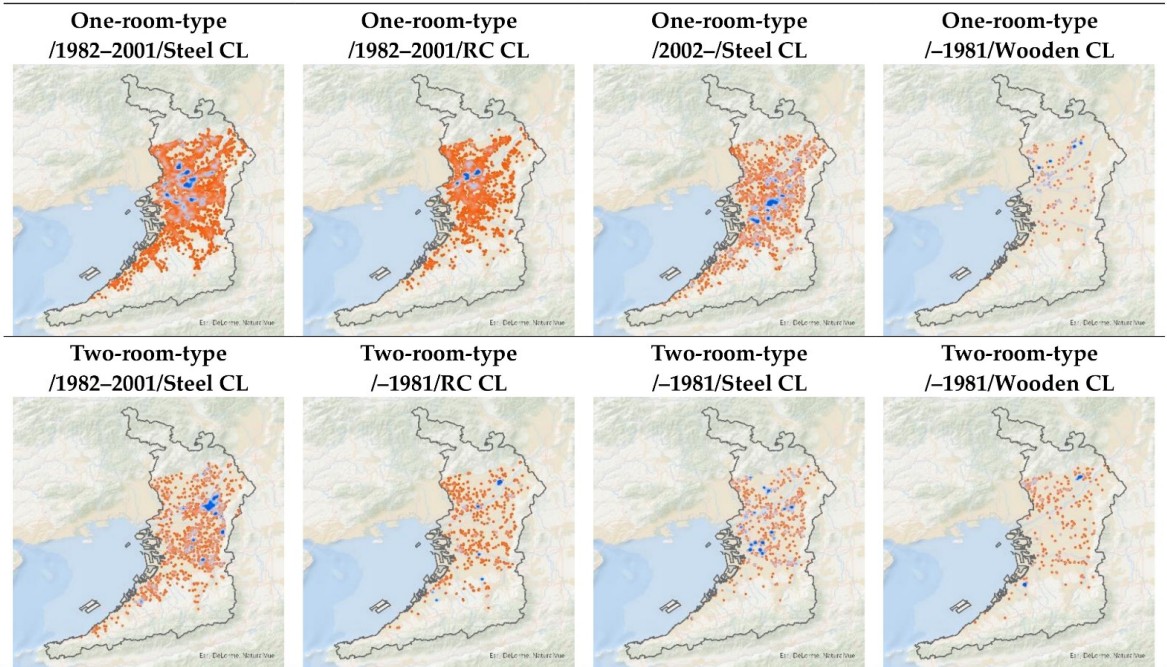

**Figure 4.** *Cont.*

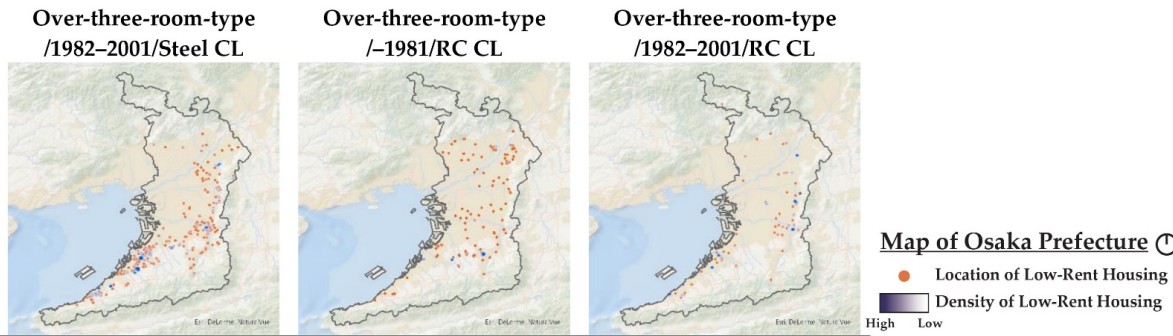

**Figure 4.** Geographic concentration of low-rent housing.

### 3.4.1. Geographic Concentration of One-Room-Type LRH

Figure 4 shows that the one-room types of LRH were distributed throughout Osaka Prefecture. In each cluster, it was found that [one-room-type/1982–2001/Steel CL] homes were clustered in several areas in the northern part of the prefecture. We also found that [one-room-type/1982–2001/RC CL] homes were accumulated in two areas in the northern part of the prefecture and that [one-room-type/2002–/Steel CL] homes were accumulated in the central part of the prefecture. Although there was a localized accumulation of [one-room-type/–1981/Wooden CL] homes in some areas, the supply of such houses was almost evenly distributed throughout Osaka Prefecture. Therefore, we found no extreme disadvantage in terms of location.

### 3.4.2. Geographic Concentration of Two-Room-Type LRH

Figure 4 shows that the two-room types of LRH were distributed throughout almost all of Osaka Prefecture, except for the southern area. In each cluster, we found that [two-room-type/1982–2001/Steel CL] homes were accumulated in the northeastern prefecture. We also found that [two-room-type/–1981/RC CL] homes were locally accumulated in the northern part of the prefecture, but no other notable trends were observed. Additionally, [two-room-type/–1981/Steel CL] homes were accumulated in some areas in the central and northeastern part of the prefecture. Lastly, [two-room-type/–1981/Wooden CL] homes were generally distributed throughout Osaka Prefecture, but showed a tendency to be accumulated in the peripheral and northern parts of Osaka Prefecture. Therefore, these homes were found to be somewhat disadvantageously located compared with one-room-type LRH.

### 3.4.3. Geographic Concentration of Over-Three-Room-Type LRH

Figure 4 shows that no significant concentration was observed for over-three-room-type LRH due to the small supply and distance between these properties. However, we found that [over-three-room-type/1982–2001/Steel CL] homes were located in the periphery of Osaka Prefecture, especially in the southern area. Therefore, such homes were located in more disadvantageous areas compared with one-room-type LRH and two-room-type LRH.

## 4. Discussion and Conclusions

This study found that housing affordability is problematic in terms of both quantity and quality among private rental apartments for multiple households in Osaka Prefecture. The novelty of this finding is essential because Osaka Prefecture has faced housing affordability issues according to the room types. These results mean that Osaka Prefecture faces the same housing affordability problems found among cities worldwide [30–33], including Tokyo [23]. Although the real estate market appears to have no problem with the supply balance of LRH, the supply volume of LRH is significantly insufficient. The results suggest a lack of LRH with multiple room types due to the imbalance in supply for each room type.

This result is significant because we found few LRH options for multi-person households; those available were found to be old, small, and unevenly located.

The supply of LRH units was distributed as follows: 88.3% were one-room types, 9.8% were two-room types, and 1.8% were over-three-room types. The total amount of two-room and over-three-room-type LRH was less than 8000 houses, accounting for only 4.2% of all private rental apartments in the real estate market. The distributed supply of LRH has potential risks to maintaining a stable life for LIGs such as disabled households, elderly people living alone, and single-parent households. These LIGs are forced to live in higher-rent housing or one-room-type LRH with a smaller GFA. These imply the social mobility issue, which indicates movements between social classes, occupational groups, and opportunities available for advancement from the sociological perspective [41]. One-room-type LRH has played a role as the first step towards independence from parents. However, the imbalance between excessive one-room-type LRH and insufficient multi-room-type LRH might have fixed the social inequality of the younger generation in terms of social mobility [42]. The social problem has caused disadvantages for the next generation of young people, and requires intervention by the government policy.

In Osaka Prefecture, old wooden LRH, such as Nagaya and Bunka, was supplied in large numbers during the postwar period and subsequently functioned as housing for LIGs [25–28]. However, we found that the role of old wooden homes as LRH has declined. Old wooden LRH units are now in minimal circulation in the real estate market and are deteriorating rapidly. In terms of location, old wooden LRH was found to be particularly concentrated in the northern part of Osaka Prefecture and widely distributed throughout Osaka Prefecture.

These housing affordability problems may be caused by imbalances between the demand from LIGs and supply from the real estate market for private rental apartments. Private real estate companies cannot profitably supply LRH to multi-person households. An essential solution to this problem could involve more generous rent subsidies by the government. This solution would allow LIGs to live in LRH with reasonable prices and living conditions. Expanding the government's coverage of rent subsidies would be effective for multi-person households using Housing Security Benefit [13] and Public Assistance for Livelihood Protection [14] because many private rent apartments are not supplied to low-income groups, but to middle- and high-income groups. For government rent subsidies, it would be adequate to use vacant units in private rental apartments under the Housing Safety Net Law [22], as there are many vacant stocks of private rental apartments which need to be renovated because they are over 20 years old. Therefore, the government needs to prepare incentives for real estate companies to renovate many LRH units for multi-person households.

One limitation of this study is that we could not conduct an analysis based on specific demand numbers for LRH because the demand for private rental housing is challenging to define, and no government data exist in Japan. For the demand-side analysis, social mobility will be a key indicator because of the widening social disparities. In future research, we intend to accurately identify the issues in housing supply by conducting data aggregation and mathematically estimating demand.

**Author Contributions:** Conceptualization, M.Y.; methodology, M.Y.; software, M.Y.; validation, M.Y. and H.K.; formal analysis, M.Y.; investigation, M.Y.; resources, M.Y. and H.K.; data curation, M.Y.; writing—original draft preparation, M.Y.; writing—review and editing, M.Y. and H.K.; visualization, M.Y.; supervision, M.Y.; project administration, H.K.; funding acquisition, H.K. All authors have read and agreed to the published version of the manuscript.

**Funding:** This research was funded by the Association of Real Estate Agents of Japan (2021).

**Institutional Review Board Statement:** Not applicable.

**Informed Consent Statement:** Not applicable.

**Data Availability Statement:** In this paper, we used the "At Home Dataset" provided by At Home Co., Ltd., via the IDR Dataset Service of National Institute of Informatics. The data presented in this study are available on request from the corresponding author, H.K.

**Acknowledgments:** We appreciate At Home Co., Ltd., and the IDR Dataset Service of National Institute of Informatics.

**Conflicts of Interest:** The authors declare no conflict of interest.

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
