# Peer review of "Housing Affordability of Private Rental Apartments According to Room Type in Osaka Prefecture"

_sustainability, doi:10.3390/su14127433_

Round 1

Reviewer 1 Report

A relevant topic is considered in the paper, and contains very interesting analysis. It has signs of scientific novelty, contains results of the analysis carried out properly. The drawn conclusions are logical and proved. The empirical results are presented visually and clear. This is a high-quality paper.
I recommend to continue the research in the field and to study housing affordability of private rental apartments in other prefectures of Japan, having collected data for the post-COVID period, since 2021 and later. As for the reviewed paper, it is high-quality.

Author Response

Dear Reviewer 1:

We appreciate the reviewer for the generous comment on the manuscript. We have attached our response letter in PDF format. We believe that the manuscript is now suitable for publication in Sustainability and look forward to hearing from you concerning your decision.

Yours sincerely

Mikio Yoshida, Haruka Kato

Reviewer 2 Report

The article analyses the affordable housing accessibility problem for the case of Osaka.

The purpose of the essay is clear, the structure and methodological criteria are adequate, and the reading of the results is correct.

The topic is very important in contemporary society, and it cuts across many geographies and disciplines, from sociology to architecture. One of the aspects that remains to be launched to broaden the context is about state policies when based on the private or public sector and the financialisation politics in the housing sector.

For a reading with a large disciplinary scope, I suggest that the cases described, in addition to the photos taken from google street, if possible, be completed with the architectural plans of the apartments in order to make qualitative associations to the quantification of the areas.

It would also be enriching to have the sociological elements about the type of population that looks for these economic houses. Among these aspects, it seems to us that the aspects of social mobility, as put by Pierre Bourdieu, can be an important indicator to understand how the One-room typology is used for generations and remains in the urban structure.

Author Response

Dear Reviewer 2:

We appreciate the reviewer for the generous comment on the manuscript. We have attached our response letter in PDF format. We believe that the manuscript is now suitable for publication in Sustainability and look forward to hearing from you concerning your decision.

Yours sincerely

Mikio Yoshida, Haruka Kato

Reviewer 3 Report

Thank you for the opportunity to read the paper "Housing Affordability of Private Rental Apartments According to Room Type in Osaka Prefecture".

Review.

- The paper touches on an important topic concerning housing affordability and access to housing in Osaka, Japan.

- I found this paper well written.

- The title is suitable for the paper and the abstract is good and clear.

- The introduction/theoretical part is in accordance with the main theme.

- The final findings and remarks are interesting. 

- I think the last sentence, about the study limitations, could be somewhere in the methodology section.

Author Response

Dear Reviewer 3:

We appreciate the reviewer for the generous comment on the manuscript. We have attached our response letter in PDF format. We believe that the manuscript is now suitable for publication in Sustainability and look forward to hearing from you concerning your decision.

Yours sincerely

Mikio Yoshida, Haruka Kato
